# Evaluation of Fixational Behavior throughout Life

**DOI:** 10.3390/brainsci12010019

**Published:** 2021-12-24

**Authors:** Irene Altemir, Adrian Alejandre, Alvaro Fanlo-Zarazaga, Marta Ortín, Teresa Pérez, Belén Masiá, Victoria Pueyo

**Affiliations:** 1Ophthalmology Department, Hospital Universitario Miguel Servet, 50009 Zaragoza, Spain; irealtemir@gmail.com (I.A.); afanloz@salud.aragon.es (A.F.-Z.); 2Instituto de Investigación Sanitaria de Aragon (IIS Aragón), Universidad de Zaragoza, 50009 Zaragoza, Spain; teyeroche@hotmail.com; 3DIVE Medical S.L., 50018 Zaragoza, Spain; adrian.alejandre@dive-medical.com (A.A.); ortin.marta@dive-medical.com (M.O.); 4Instituto Universitario de Investigación de Ingeniería en Aragón (I3A), Universidad de Zaragoza, 50009 Zaragoza, Spain; bmasia@unizar.es

**Keywords:** eye tracking, gaze stability, visual fixation, visual development

## Abstract

Background: To quantify development of gaze stability throughout life during short and long fixational tasks using eye tracking technology. Methods: Two hundred and fifty-nine participants aged between 5 months and 77 years were recruited along the study. All participants underwent a complete ophthalmological assessment. Fixational behavior during long and short fixational tasks was analyzed using a DIVE (Device for an Integral Visual Examination), a digital test assisted with eye tracking technology. The participants were divided into ten groups according to their age. Group 1, 0–2 years; group 2, 2–5 years; group 3, 5–10 years; group 4, 10–20 years; group 5, 20–30 years; group 6, 30–40 years; group 7, 40–50 years; group 8, 50–60 years; group 9, 60–70 years; and group 10, over 70 years. Results: Gaze stability, assessed by logBCEA (log-transformed bivariate contour ellipse area), improved with age from 5 months to 30 years (1.27 vs. 0.57 deg^2^ for long fixational task, 0.73 vs. −0.04 deg^2^ for short fixational task), while fixations tend to be longer (1.95 vs. 2.80 msec for long fixational tasks and 0.80 vs. 1.71 msec for short fixational tasks). All fixational outcomes worsened progressively from the fifth decade of life. Log-transformed bivariate contour ellipse area (0.79, 0.83, 0.91, 1.42 deg^2^ for long fixational task and 0.01, 0.18, 0.28, 0.44 deg^2^ for short fixational task, for group 7, 8, 9, and 10 respectively). Stimuli features may influence oculomotor performance, with smaller stimuli providing prolonged fixations. Conclusions: Fixational behavior can be accurately assessed from 5 months of age using a DIVE. We report normative data of gaze stability and duration of fixations for every age group. Currently available technology may increase the accuracy of our visual assessments at any age.

## 1. Introduction

Visual fixation, defined as the maintenance of the gaze on a certain point, is essential for the correct development of the visual function [1]. Visual function includes several aspects, such as visual acuity (VA), contrast sensitivity (CS), color vision, stereopsis, and oculomotor control.

Different methods have been proposed to measure visual fixation. The most widespread in clinical practice is the subjective observation of the participants fixation, on an object requested to look at and follow [2,3]. In general, with this type of evaluation, the fixation can be described as central, stable, and maintained, or as eccentric, unstable, and not maintained. It is necessary to look for a more objective technique that is not so dependent on the skills of the observer. There are several technologies able to assess gaze position, such as electro-oculography [4,5], confocal laser scanning (SLO) [6], and eye tracking [7,8]. Among all of them, eye-tracking seems to provide the highest accuracy in a non-invasive way [6,9,10]. Eye tracking technology uses the vector between the center of the pupil and the corneal reflection created by an infrared light to calculate the position of the gaze on the screen.

At birth, fixation is not fully developed, and it is during the first years of life when it matures until adulthood, while the fovea and the central nervous system (CNS) reach their adult structure and function. Unstable fixation in children can be due to different pathologies, such as congenital cataract or strabismus [11,12]. Larger fixations and instability were documented in children with strabismus using eye-tracking technology [13]. Amblyopia is frequently associated with irregular eye movements and poor fixation [14], while children with cerebral visual impairment (CVI) exhibit, among other visual and oculomotor dysfunctions, instability of fixation [15]. Eye movements in dyslexic children have been studied as well, showing that they spent more time reading the text and had longer duration of the fixations [16]. It has also been studied whether decrease in vision could be related to poor gaze stability. An unstable gaze implies a poor visual function, with poor binocular vision and stereopsis [17].

In adult patients, fixation has been used to evaluate visual performance in certain ocular pathologies, such as age-related macular degeneration (AMD) [18], central nervous system disorders [19], cataract [20], or amblyopia [21]. Positive correlation between average displacement of the fixation point, and visual acuity decrease for all types of amblyopia were found [21]. Gonzalez et al. found that patients with amblyopia exhibited a significant decrease in fixation stability using bivariate contour ellipse area (BCEA) [22].

Increased intrusive saccades during fixation in adults have also been described in brain injuries or ataxia [23,24]. Assessment of oculomotor control has, therefore, been proposed as an objective and quantitative outcome for monitoring certain visual and neurological disorders.

The evaluation of visual fixation or visual pattern in healthy patients has been studied by several authors, using different evaluation techniques [6,10,25,26,27,28,29,30,31,32,33]. Although all of them share the goal of providing insights into the development of visual fixation throughout life, relevant differences in the methodology of the performed studies make comparisons among them challenging. For example, Galambos et al. [34] studied fixation on half-face stimuli, while other studies used the standard stimuli presented by Macular Integrity Assessment (MAIA) or MP1 microperimeters. As far as the eye tracker, there are many differences among them, such as sampling rate (ranging from 25 to 250 Hz) duration of the stimuli or accuracy. Different methodologies give rise to inconsistent oculomotor outcomes in healthy participants and patients with visual disorders. Precise description of normal outcomes for every age with every technology and methodology is essential before using oculomotor control in clinical practice.

To the best of our knowledge, there are no previous studies reporting physiological fixational performance throughout life. This study bridges the important gap between basic research and clinical care. The aim of our study was to provide reference values for fixational skills in healthy participants, from 5 months to 75 years, and quantify gaze stability and fixational outcomes during short and long fixational tasks.

## 2. Materials and Methods

### 2.1. Participants

Two hundred and fifty-nine participants were evaluated for this study. All of them were born at term (>37 weeks of gestational age), with no known ocular disease (except low ametropia), and no neurological or systemic disorder. Participants with strabismus were excluded. All participants with refractive errors equal or higher than ±1.5 diopters of sphere in adults (more than 14 years old) or equal or higher than ±2.5 sphere in younger than 14 years, or equal or higher than 1.5 diopters of astigmatism, were excluded. In addition, all of them were verified that the no corrected visual acuity (VA) was optimal (0.8 Snellen or 0.1 logMar or more) at 50 cm, for a correct performance of the test. They were recruited from healthy parents or siblings of participants, and family members of the employees of the Department of Ophthalmology. According to their age, the participants were divided into ten groups: group 1, 0–2 years; group 2, 2–5 years; group 3, 5–10 years; group 4, 10–20 years; group 5, 20–30 years; group 6, 30–40 years; group 7, 40–50 years; group 8, 50–60 years; group 9, 60–70 years; and group 10, over 70 years.

To take part in the project, written informed consent was obtained from all adults, or parents or guardians of children. This study respects the tenets of the Declaration of Helsinki and was approved by a local ethics committee (CEICA, Comité Ético de Investigación Clínica de Aragón) (PI15/0157).

### 2.2. Examination

All participants underwent an ophthalmologic evaluation: best-corrected visual acuity, ocular motility, stereoacuity with TNO, refraction under cycloplegia, slit lamp examination, Goldmann tonometry (participants older than 16 years old), indirect funduscopy and visual fixational behavior evaluation. VA was assessed in cooperative participants monocularly and binocularly, with optotypes adapted to each participant’s age based on commercially available LEA symbols or letters at 3-m distance. Grating acuity was obtained using the LEA paddles following a preferential looking paradigm from all infants younger than 24 months of age, and those non-cooperatives for the previously detailed VA assessments.

### 2.3. Fixational Behavior Assessment

The examination of fixational behavior was carried out using a prototype of DIVE (Device for an Integral Visual Examination) with a 12-inch high resolution screen, corresponding to a visual angle of 28.46 deg horizontally and 19.19 deg vertically. DIVE is a device composed of a high-resolution screen and eye tracking technology. Its software includes several visual tests (VA, contrast sensitivity, color perception, and oculomotor control), which can be performed in any person from some months of age, with very low need of cooperation from the participants. All the tests can be consecutively performed in a matter of minutes. They only sit in front of the screen and the device automatically provides all the instructions to the participant. They are encouraged to keep quiet and steady looking at the different visual stimuli presented on the screen. The instructions were the same for everyone: adults, children, and pre-school children. When it comes to a young child, the same instructions are given to their parents to keep their heads steady.

All eye movements were recorded during the test using eye tracking technology integrated in the DIVE below the screen, with a maximum temporal resolution of 60 Hz. The average binocular accuracy of the eye tracker is around 0.5 to 1° of visual angle.

The eye tracker used in this study is from The Eye Tribe. It is non-invasive, supports free-head movements, works at 60 Hz, and has an accuracy of 0.5–1° [35].

A controlled lighting source was also included consisting of a small LED bar that, connected to a USB port, generates a uniform illumination of 6000 K and a luminance of 250 lumens.

The test was performed with all the lights off, except for the light incorporated in the device, and without any distractors on the walls.

First, a calibration procedure of the eye tracker was performed for each participant to guarantee accuracy and precision of gaze tracking. The calibration consisted of an image of an animal that appears and disappears in nine points uniformly distributed across the screen. All the participants were requested to look at the images, keeping their head and body steady. Children younger than 24 months were positioned on a parent’s lap, who was asked to keep their heads steady. The test is carried out 50 cm from the screen (Figure 1); this distance is always checked before starting the calibration and performing the test.

A calibration was considered to be good quality if the metrics were lower than an empirically set threshold. The study included two different fixational tests in order to evaluate fixational behavior in conditions similar to daily life activities. Children are used to have screens and they are an everyday tool in every home today. The first one presented a long fixational task. It consisted of a high-contrast cartoon of a child of 3 deg × 1.56 deg appearing on the center of the screen, who talked to the participant for 10 s. Only the central nine seconds were taken into account for the study. For the short fixation tasks, we included data from the 2 central seconds of each trial.

During the second part of the exam, short fixational tasks were presented. The fixation target consisted of a stylized image of a bee. This exposure lasts 3 s, but only the two central ones are taken into account for the study (Figure 2).

Three sequences of eight stimuli each were consecutively presented. Stimuli have a combination of three characteristics: size (8 mm or 17 mm), sound (with sound or without sound), and movement (static or small buzz simulating the vibration of the bee). All possible combinations of those three characteristics were used to define the eight stimuli of each sequence, which were then presented randomly. Each sequence begins with a central fixation stimulus, which moves to another random peripheral position every 3 s, with no overlapping and a fixed distance of 7.74 degrees between consecutive stimuli.

The fixational parameters analyzed both during long and short fixational tasks were gaze stability and median duration of fixation (DF). To quantify the gaze stability, the bivariate contour ellipse area (BCEA) was calculated. It refers to the area in squared degrees (deg^2^) of the ellipse around the stimulus containing most of the fixation points Therefore, the smaller the BCEA value, the more stable the gaze is. The BCEA covering 68.2% of the fixation points was calculated using the following equation:

BCEA = 2 ∗ k ∗ π ∗ σx ∗ σy ∗ (1 − p2) 1/2

where,

σx = standard deviation of the horizontal position of the eye,

σy = standard deviation of the vertical position of the eye,

2.291 is the value χ2 (2 df) that corresponds to a probability of 0.68, and p is the correlation coefficient of the moment of the Pearson product of the horizontal and vertical ocular positions.

Since the BCEAs are usually not normally distributed, we used a natural log transformation on their values to normalize data (logBCEA).

We studied gaze stability during the fixed period of time when the participant was asked to fixate on a target, which implies that we assume we are including within- and between-fixations data. For the long fixation task, the 9 central seconds of the trial were included in the study. For the short fixation tasks, we included data from the 2 central seconds of each trial.

### 2.4. Statistical Analysis

All data were analyzed using SPSS 25.0 (SPSS Inc., Chicago, IL, USA). Descriptive characteristics were described by the mean, standard deviation, and ranges, while the results of the fixation were described by the median and interquartile range, due to the dispersion of the data.

The groups were compared by analysis of variance (ANOVA) or by the Kruskal–Wallis test. Multivariate analyses were performed including gender, age, VA, and refractive error as independent variables, and the results of fixational assessments as dependent variables. We compared the oculomotor behavior between the different age groups and evaluated the influence of the size, sound, and vibration of the visual stimuli on the results of the fixation.

## 3. Results

A total of 259 participants were included in the study. One hundred and forty-six were female (56.4%) and 113 were male (43.6%). Average age of the participants was 19.60 years (SD 21.12 years, range 0.43–77.07 years). Best corrected binocular VA was −0.00 ± 0.05 in logMAR scale (6/6 in Snellen scale). Mean refraction of the study group was 0.40 ± 0.73 of sphere and 0.00 ± 0.28 of cylinder for the right eye, and 0.48 ± 0.72 of sphere and 0.10 ± 0.28 cylinder for the left eye. According to their age, the participants were divided into ten groups: 40 participants in group 1 (between 0 and 2 years), 52 participants in group 2 (between 2 and 5), 37 participants in group 3 (between 5 and 10 years), 39 participants in group 4 (between 10 and 20 years), 24 participants in group 5 (between 20 and 30 years), 13 participants in group 6 (between 30 and 40 years), 17 participants in group 7 (between 40 and 50 years), 18 participants in group 8 (between 50 and 60 years), 14 participants in group 9 (between 60 and 70 years), and 5 participants in group ten (over 70 years). All of the included participants presented normal ocular motility and eye fundus. Descriptive data and visual results are given in Table 1. Although visual function was within normal ranges in all the cases, visual and refractive parameters slightly differed between groups. There was a tendency to worse VA in the participants from the oldest age groups.

Fixational outcomes in each group, during short and long fixational tasks, are reported in Table 2. Statistical differences were found between age groups, since fixational behavior differed significantly throughout life. There is a clear tendency to worse gaze stability and decreased duration of fixations during the first 5 years and after the fifth decade of life.

Regarding the influence of the stimuli features, smaller stimuli seem to provide longer fixations, while those with a small vibration were related to more stable fixations. However, no influence of sound was observed on any fixational parameter (Table 3). However, this influence was greater in the youngest participants. Participants younger than 10 years were more susceptible to small movement and older ones to its size. In the multivariate analysis including age and gender as independent variables, gender was not included in the model (*p* > 0.05), but age showed influence on fixation stability in long and short fixational tasks (Beta = −0.149, *p* = 0.029; and Beta = −0.201, *p* = 0.002, respectively). However, when we adjusted the model for VA, spheric equivalent and cylinder defects age was excluded from all the models (*p* > 0.05). No influence of age was observed on the duration of fixations both in adjusted and unadjusted models (Figure 3).

## 4. Discussion

Oculomotor performance is mostly assessed by clinical observation using an attractive visual stimulus [2,3]. It is a subjective and imprecise examination, which can only identify severe disorders. In this work, we have described the normal fixational behavior throughout life, examining a group of participants (from 5 months to 77 years) in a precise, objective, and simple way, using a very novel technology.

To carry out this study we have used an Eye Tribe eye tracker. This instrument works at 60 Hz, and with an accuracy of 0.5–1°. It is non-invasive and supports free-head movements [35]. For reference, the IS5 eye tracker with similar characteristics by the company Tobii, one of the most widely used eye tracking companies, reports an accuracy of 1.71° and a precision of 0.79°, also with a frequency of 60 Hz [36]. Invasive eye tracking devices incorporated into glasses are not usable by non-collaborative patients and are meant to track the gaze in 3D environments, so they do not report comparable accuracy values. Virtual reality headsets, also to be used by collaborative patients, may include eye tracking as well. As an example, VIVE Pro Eye headset reports an accuracy of 0.5–1.1° with a frequency of 120 Hz [37].

We therefore believe that our metrics can be comparable to those we would obtain with other eye tracking technologies, and that our conclusions would still stand no matter the eye tracker employed to perform the experiments. A more detailed comparison of the metrics reported in this paper with different eye tracking systems is left for future work.

Despite our eye-tracker sampling rate being lower than the one from other eye-trackers, it is enough to accurately study fixational performance [38]. This frequency has been used in several studies [39,40]. We have presented how fixational behavior seems to improve mostly during the first five years of life but keeps on stabilizing until the third decade, either during long fixational and short fixational tasks. Along third and fourth decades mean duration of fixations and logBCEA exhibit a flat curve, as presented in Figure 3. Beyond this point, fixation stability seems to worsen, with a mild slope in the curve during the fifth, sixth, and seventh decades and a more pronounced slope from the eighth decade. This tendency can be observed both during long and short fixational tasks. However, the influence of age in the duration of the fixations is much subtler.

Finding worse fixation stability results in the first years of life is in agreement with the evidence that the ability to steadily fixate is not fully developed at birth. It requires retinal maturation and accurate control from the CNS to allow and maintain steady fixation.

In addition, the evaluation of oculomotor control outcomes must take into account the influence of certain cognitive processes such as attention, information processing, memory and anticipation, on fixational behavior and also on later intellectual function in childhood [41]. On the other hand, worsening fixational behavior from the fifth decade of life, may be due to several factors, such as opacification of the lens [20] or macular diseases, such as age-related macular degeneration [42].

Age was included in regression models as an independent factor, while gender was not. However, none of them were included in models adjusted for VA, spheric equivalent, and cylinder defects. It might be due to the non-linear relationship between age and fixational performance, but also to potential causes of this relationship, which could be influenced by intermediate factors, such as refractive error or VA.

The influence of age on oculomotor control has been previously studied with inconsistent results. Smooth pursuit movements seem to slow with age [25], while the range of voluntary eye movements becomes restricted [26]. Differences in the pattern of stability along the horizontal and vertical axis have been found between old and young participants, with older ones showing greater horizontal variability [27]. However, fixational behavior might be slightly protected versus other functions due to differences in the susceptibility of extraocular muscle fiber types against age, since small fibers—which are more involved in the tonic control of fixation—are less affected [43,44]. Other authors have studied the influence of age in visual fixation. Molina-Martín et al. [31] carried out a study of 237 eyes of 237 heathy subjects aged between 10 and 70 years, using MAIA microperimeter. They found that the stability of the pattern of fixation in healthy eyes tends to decrease with age. The group of Fragiotta studied 85 eyes without ophthalmic disorders using microperimeter, and they concluded that aging increases BCEA [32].

Different instruments and proceedings for measuring visual fixation have also been studied in healthy participants. Dunbar et al. [33] evaluated sixteen healthy participants. In this work, fixation stability was recorded monocularly on the SLO and the MP-1. They did not find significant differences between BCEA values from the SLO and MP-1 in healthy participants. Liu et al. [6], using two methods for the evaluation of fixation, found the same results. They evaluated 41 normal eyes using MP-1 and OCT/SLO. BCEA was 2.93 ± 0.32 log minarc^2^ on OCT/SLO and 2.89 ± 0.30 log minarc^2^ on MP-1 [9]. Morales et al. [30] described a study formed by 358 participants, in which MAIA microparameter was used. Visual fixation was assessed using the BCEA. They found areas of 2.40 deg^2^. This value is quite similar to our findings for long fixational tasks. They posited that the most precise parameter to be used for the evaluation of fixation was the BCEA at 95%. Gonzalez et al. (2012) carried out a study with adults for 15-s intervals using an EyeLink 1000 and found log BCEAs of −0.88 log (°^2^) for binocular viewing, similar to our values in long fixational task [22].

Aring et al. studied fixational behavior in a group of children aged from 4 to 15 years. They found that the fixation density is more centered around the center of gravity the older the child is, and fixation time increases with age, while intruding saccades decrease with increasing age [28,29]. They used an infrared-tracking device. Their results are in agreement with those found in our study.

Gaze stability in 4-to-10-week-old infants was evaluated by Seemiller et al. [45], who reported that gaze stability was similar in infants and adults. However, calibration procedure was only performed in adult participants, while all the tests performed in infants were calibrated by an adult. Furthermore, gaze data were analyzed using adult criteria, such as removing adult-like saccades from infant trials. After undergoing a calibration in every participant, we applied customized algorithms to the collected raw gaze data to recover sporadic lost samples with interpolation and filter out the nose created by eye jittering, which could significantly alter our metrics. We consider that lack of accurate calibration could be an important source of bias. Krista K, et al. carried out a study with 160 children (aged 4–12 years) with anisometropia and/or strabismic and compared them with 46 age-matched controls. They used a binocular eye-tracker (EyeLink) and found worse BCEA for children with anisometropia and/or strabismus [13].

The goal of this study was to assess gaze stability in a large sample of healthy participants throughout life from only 5 months of age. We are therefore assuming that we include in the analysis both within and between fixations data. However, assessing gaze stability rather than fixation stability provides, in this case, a better description of fixational behavior in real settings, and allows a more accurate comparison between ages.

Fixation is the basis of oculomotor control. A good stability of fixation is a prerequisite for an optimal visual function. Unstable visual fixation has been linked to many visual disorders.

Children with amblyopia exhibit worse oculomotor control in amblyopic eyes compared with fellow eyes, while fellow eyes show even better fixation stability than eyes from non-amblyopic children [46]. On the other hand, Chung S et al. [14] defined and described the major problems in ocular movements in their study. They found that, in general, the characteristics of fixational eye movements are not significantly different between the fellow eyes of amblyopes and control, and that the strabismic amblyopic eyes are always different from the others. Subramanian et al. [17] analyzed 89 children with the Nidek MP-1 perimeter and reported significantly larger ellipse areas for amblyopic eyes than for the fellow eyes and the eyes of normal controls, with worse stereoacuity, not only in the amblyopic eye, but also in the contralateral eye. In these participants, longer duration of decorrelated visual experience is associated with increased fixation instability, poorer stereoacuity, and more severe amblyopia. Thus, fixation stability has shown a good correlation with VA and stereoacuity in patients with amblyopia [17]. It suggests that it could be an indicator of the loss of visual function.

Fixation stability is also reduced in people with central visual loss and extrafoveal locus of fixation [47]. A direct correlation has also been found between steady fixation and VA or reading speed in patients with macular disease [48]. Improving fixation stability is therefore the goal of most training programs in macular disease [49]. Macular pathologies such as AMD give rise to poor fixation. In the study carried out by Sivaprasad et al. [18], fixation was analyzed in 77 patients with wet AMD who were being treated with ranibizumab. Microperimetry was used to evaluate visual function and fixational skills, and a relationship between visual fixation and VA was found, the latter being higher in the eyes with central and stable fixation. On the other hand, no relationship between the optical coherence tomography measurements and fixational parameters was found. However, Rohrscheneider K et al. [50] analyzed 30 eyes with AMD and 20 with juvenile macular dystrophy and did not find a relationship between the VA value and fixation stability. The analysis in this case was carried out with a SLO and the fixation stability was more related to morphological changes in the macula than to visual function. The measurement of fixation stability by the BCEA has been extended in this field. Fixation stability has also been used for assessing clinical improvement after surgery for a macular hole. Tarita-Nistor L, et al. found shorter BCEA values (0.35 deg^2^ versus 0.29 deg^2^) after a macular hole surgery using a MP-1 microperimeter showed more stable fixation [51].

Visual fixation is essential for a good quality of life due to its relationship with daily life tasks, such as reading or driving. Crossland MD et al. [52] analyzed 25 patients with AMD and with juvenile macular dystrophies and found a relationship between reading speed and fixation stability, but they did not find any relationship between fixation and VA, scotoma size, or sensitivity to contrast. In order to monitor eye movements, they used a SLO and an infrared light [10].

Fixational skills are mostly subjectively assessed by clinical observation. Nowadays, available technology may provide more accurate and objective assessment of oculomotor skills. Many instruments can be used to measure ocular movements, such as SLO, electro-oculography (EOG), scleral search coil, video-oculography (VOG), or eye trackers [53]. Eye tracking seems to be the most recommended tool in recent studies as it provides more precise measurements in a non-invasive way [3,14]. The sampling frequency varies a lot among the different eye trackers used in the clinical and investigation practice; the one used in our study is high enough to describe fixations in detail. In our study, the evaluation was done binocularly, since it better reflects fixational skills in real life and can be easily performed, even in very young children [54].

Gaze or fixation stability can be quantified using different methods, such as the Bivariate Contour Ellipse Area (BCEA) [55] or the Centre of Gravity [28]. BCEA has been widely used as a useful parameter and can be calculated both for ET and for SLO data. The BCEA is a very useful tool for visualizing a participant’s fixation results. It provides an intuitive visual representation, plotting an ellipse containing fixation positions registered during the measurement procedure.

A secondary objective of our study was to see if any of the features of the stimulus could have any influence on the results. We did not find a significant impact of sound on the fixational outcomes, but we found longer fixations with smaller stimuli and more stable fixations with small vibration. Worse fixation and larger areas of the ellipse with larger stimuli has already been reported by other authors [54], although they also presented stimuli with different shapes (a point, a cross, and a ring) apart from different sizes. At this moment, a “gold standard” fixation test has not been well established (with a specific size and form). This fixation test could be better used in daily evaluation of visual function [56].

The main limitations of the study are those related to the number of participants in some study groups, mainly due to difficulties finding old participants meeting all the inclusion criteria. Another limitation of the study may be the difficulty of obtaining reliable tests in younger children.

Current technology can significantly improve the assessment of oculomotor performance. However, before using it in clinical practice, normative data should be provided. We report a description of fixational skills throughout life during tasks usually performed in daily life.

In conclusion, our study shows normative data in healthy participants with values of fixation performance. This can be useful for the early diagnosis or follow-up in some pathologies in which fixation and eye movements are affected.

In order for a device assessing oculomotor control, as DIVE, to be used in daily clinical practice, it is essential to have normative data by age for fixational behavior.

## Figures and Tables

**Figure 1 brainsci-12-00019-f001:**
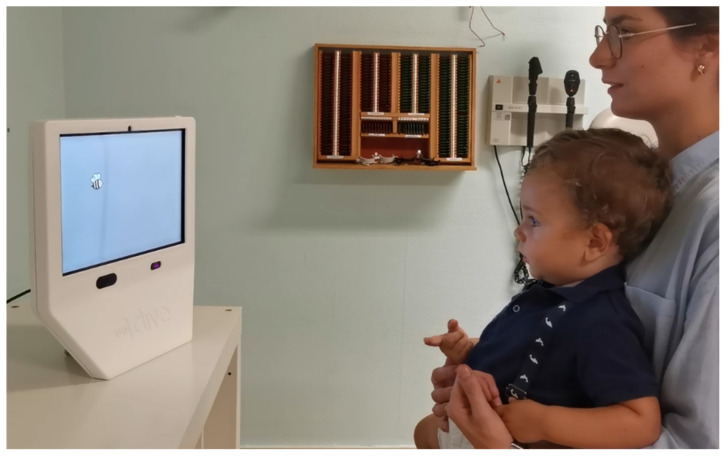
Representation of the device.

**Figure 2 brainsci-12-00019-f002:**
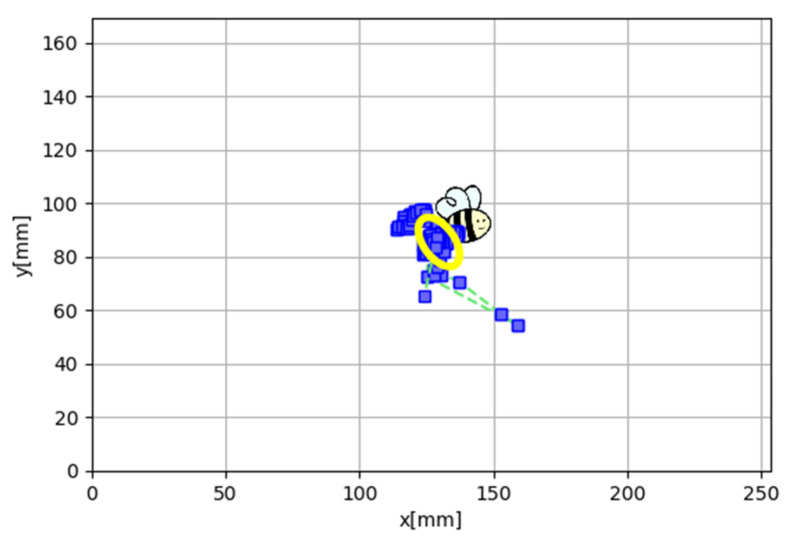
Representation of the fixation on visual stimulus of one study participant with the plot of the bivariate contour ellipse area (BCEA).

**Figure 3 brainsci-12-00019-f003:**
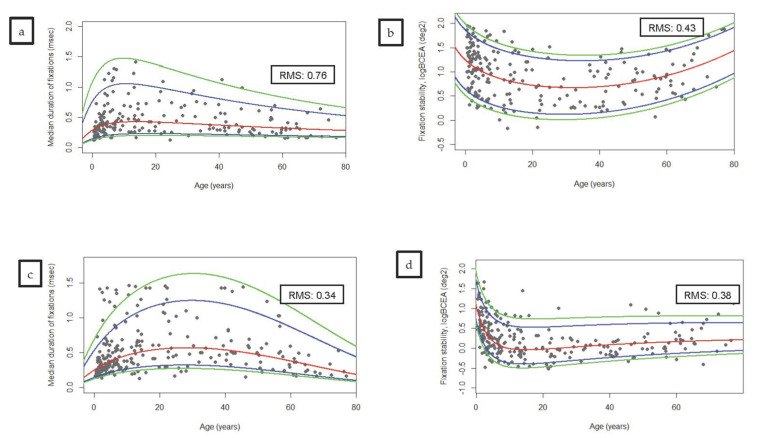
Scatterplots representing (**a**) median duration of fixation with long fixational task, (**b**) fixation stability with long fixational task, (**c**) median duration of fixation with short fixational task, (**d**) fixation stability with short fixational task, outcomes plotted against age. Legend: From bottom to top, the first green line corresponds to P05, the blue one to P10, the red line corresponds to the one that best fits the model (P50), the second blue line to the P90, and the second green line to P95.

**Table 1 brainsci-12-00019-t001:** Demographic and visual data of the sample population.

	1 (0–2 Year)	2 (2–5 Year)	3 (5–10 Year)	4 (10–20 Year)	5 (20–30 Year)	6 (30–40 Year)	7 (40–50 Year)	8 (50–60 Year)	9 (60–70 Year)	10 (>70 Year)	Global
VA (LogMar)	-	0.03 (0.06)	−0.01 (0.04)	−0.03 (0.05)	−0.01 (0.03)	0.0 (0.02)	0.03 (0.07)	−0.00 (0.02)	0.01 (0.02)	0.03 (0.06)	−0.00 (0.05)
VA (cpd)	4.85 (2.28)	10.0 (2.83)	-	-	-	-	-	-	-	-	-
Sphere RE	2.13 (0.53)	1.03 (0.48)	1.24 (0.99)	0.16 (0.52)	−0.20 (0.51)	−0.15 (0.51)	−0.06 (0.49)	0.33 (0.97)	0.45 (0.70)	0.69 (0.55)	0.44 (0.87)
Sphere LE	2.13 (0.53)	1.14 (0.49)	1.20 (0.83)	0.30 (0.55)	−0.22 (0.56)	0.19 (0.45)	0.28 (0.63)	0.20 (0.86)	0.47 (0.59)	1.0 (0.57)	0.50 (0.82)
Cylinder RE	0.75 (0.35)	0.44 (0.39)	0.40 (0.33)	0.27 (0.33)	0.15 (0.24)	0.50 (0.18)	0.52 (0.26)	0.55 (0.20)	0.54 (0.27)	0.63 (0.25)	0.42 (0.31)
Cylinder LE	0.75 (0.35)	0.44 (0.44)	0.33 (0.29)	0.30 (0.44)	0.17 (0.25)	0.35 (0.24)	0.45 (0.26)	0.55 (0.16)	0.56 (0.29)	1.00 (0.25)	0.41 (0.34)

VA, visual acuity; RE, right eye; LE, left eye. Data are presented as mean (standard deviation).

**Table 2 brainsci-12-00019-t002:** Comparison of fixational parameters among the study groups. Data are reported as mean (95% confidence interval).

	1 (0–2 Year)	2 (2–5 Year)	3 (5–10 Year)	4 (10–20 Year)	5 (20–30 Year)	6 (30–40 Year)	7 (40–50 Year)	8 (50–60 Year)	9 (60–70 Year)	10 (>70 Year)	*p*
Long Fixational Tasks											
Median duration of fixation, s	0.25 (0.21–0.29)	0.43 (0.36–0.49)	0.58 (0.43–0.73)	1.10 (0.51–1.69)	0.81 (0.24–1.38)	0.53 (0.36–0.63)	0.60 (0.38–0.68)	0.37 (0.28–0.45)	0.29 (0.22–0.36)	0.37 (−0.00–0.73)	0.001
Ellipse area (logBCEA) deg^2^	1.27 (1.14–1.41)	1.04 (0.90–1.18)	1.05 (0.85–1.25)	0.77 (0.57–0.97)	0.57 (0.33–0.81)	0.56 (0.35–0.77)	0.79 (0.56–1.01)	0.83 (0.66–1.00)	0.91 (0.70–1.13)	1.42 (0.70–2.14)	<0.001
Short Fixational Tasks											
Median duration of fixation, s	0.41 (0.32–0.50)	0.59 (0.51–0.68)	0.49 (0.40–0.58)	0.67 (0.57–0.76)	0.82 (0.67–0.96)	0.84 (0.66–1.00)	0.80 (0.65–0.96)	0.70 (0.56–0.84)	0.51 (0.32–0.70)	0.42 (0.55–0.79)	<0.001
Ellipse area (logBCEA) deg^2^	0.73 (0.56–0.89)	0.38( 0.25–0.50)	0.23 (0.11–0.35)	0.09 (−0.04–0.23)	−0.04 (−0.18–0.10)	−0.01 (−0.09–0.08)	0.01 (−0.15–0.18)	0.18 (0.01–0.37)	0.28 (0.11–0.45)	0.44 (−0.21–1.09)	<0.001

**Table 3 brainsci-12-00019-t003:** Effect of stimuli features on oculomotor outcomes.

	Target Size		Target Movement		Target Sound		p_1_	p_2_	p_3_
	Small	Large	No Vibration	Vibration	No Sound	With Sound			
Duration of fixations, s	0.70	0.64	0.67	0.67	0.67	0.66	<0.001	0.608	0.594
Fixation stability (logBCEA) deg^2^	0.32	0.32	0.34	0.29	0.31	0.32	0.958	0.039	0.606

p_1_, comparison between stimuli with two different sizes; p_2_, comparison between stimuli with or without small vibration.; p_3_, comparison between stimuli with and without sound.

## Data Availability

Data available on request due to privacy restrictions. The data presented in this study are available on request from the corresponding author. The data are not publicly available due to privacy restrictions.

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
