# Peer review of "Evaluation of Fixational Behavior throughout Life"

_brainsci, 2021, doi:10.3390/brainsci12010019_

Round 1
Reviewer 1 Report
Very well set and described research.
Authors provide reference values for fixational skills in healthy patients (normative data of gaze stability and duration of fixations for every age group) which is a great contribution to the field.
In the future authors can include patients with ocular and brain pathology and compare results which will be also a great contribution.
Author Response
The comments of the reviewers have been taken into account and have been addressed in revised version
Point 1: In the future authors can include patients with ocular and brain pathology and compare results which will be also a great contribution.
Response 1: Thank you very much for the advice, we will take it into account. It could be very interesting to include patients of this type in future work.

Reviewer 2 Report
This paper reports an assessment of a new clinical tool, an eye tracker to assess visual fixation to targets. A virtue of the work is that it spans a wide age group, from infancy to adulthood. The methods and analyses seem appropriate, and the results are clearly presented.
Despite these positives, I felt that the work was poorly located within the vast experimental literature on visual fixation in infants, children and adults. In introducing the study, little contact is made with that literature. And although some experimental findings are alluded to in the discussion, analysis is very much at a surface level with little by way of detailed comparison to the present results. Additionally, it would appear essential to compare the performance of the present eye tracker with others widely used in the experimental literature. How does the system compare with the sophisticated head mounted systems used with adults, and with the remote systems predominantly used with infants (historically, ASL systems, but currently predominantly Tobii). In addition, are there specific practical advantages of this system over some of the compact and very portable systems produced commercially by Tobii and others?
Another missing element is discussion of the limitations of eye trackers when used with infants. Infants cannot be physically restrained and frequent head movements lead to data loss. Additionally, watery eyes, frequent in young infants, make it difficult to gain useable data. All this information would seem important in assessing a tool aimed at being useful across the lifespan.
Finally, a small point. As an experimental psychologist I found it peculiar to refer to participants as patients, when they had no condition to be treated.
Author Response
The comments of the reviewers have been taken into account and have been addressed in the revised version
Point 1: Despite these positives, I felt that the work was poorly located within the vast experimental literature on visual fixation in infants, children and adults. In introducing the study, little contact is made with that literature. And although some experimental findings are alluded to in the discussion, analysis is very much at a surface level with little by way of detailed comparison to the present results.
Response 1: The introduction has been expanded and specified in the manner requested, with more literature on visual fixation. More references have been added to deepen the analysis of visual fixation in the discussion.The changes have been highlighted in red in the introduction and discussion sections.
Point 2:Additionally, it would appear essential to compare the performance of the present eye tracker with others widely used in the experimental literature. How does the system compare with the sophisticated head mounted systems used with adults, and with the remote systems predominantly used with infants (historically, ASL systems, but currently predominantly Tobii). In addition, are there specific practical advantages of this system over some of the compact and very portable systems produced commercially by Tobii and others?
Response 2: Thank you very much for the clarification and I have to agree with you. A paragraph has been added to clarify and compare our device with other eye trackers used in literature in methods and in discussion section. The DIVE device is not comparable to other compact and portable eye tracking systems, because it does not only perform eye tracking, but it also includes tests to evaluate the visual function and automatically performs an analysis of the gaze data and extract relevant metrics.
Point 3:Another missing element is discussion of the limitations of eye trackers when used with infants. Infants cannot be physically restrained and frequent head movements lead to data loss. Additionally, watery eyes, frequent in young infants, make it difficult to gain useable data. All this information would seem important in assessing a tool aimed at being useful across the lifespan.
Response 3: We agree with the reviewer that this point needs to be clarified. A sentenced has been included in the Discussion Section. To try to minimize this limitation, the parents were asked to gently hold the head and if movement was observed on the part of the child during the trial, the examination was repeated or even the patient were excluded. Even so, we assume that the immobility of young children may be a limitation for the correct performance of the test.
Point 4: Finally, a small point. As an experimental psychologist I found it peculiar to refer to participants as patients, when they had no condition to be treated.
Response 4: Thank you for your advice, we totally agree with the reviewer. We are usually biased by our clinical background. The word patient has been changed by participants in all the manuscript sections.
Each of the numbered items has been responded in this letter. The section can be seeing in this letter as well and a red headlight in the revised version.

Round 2
Reviewer 2 Report
This version is much improved in some respects. However, I still feel that there insufficient review of existing experimental data on visual fixation across the age range. This would seem essential in order to evaluate not just whether the device works, but whether it provides data that agree with experimental findings. The small amount of additional literature referred to is largely clinical, relating to special populations. In contrast, the present data are from typical individuals without pathology.
Author Response
The comments of the reviewers have been taken into account and have been addressed in the revised version
Point 1: This version is much improved in some respects. However, I still feel that there insufficient review of existing experimental data on visual fixation across the age range. This would seem essential in order to evaluate not just whether the device works, but whether it provides data that agree with experimental findings. The small amount of additional literature referred to is largely clinical, relating to special populations. In contrast, the present data are from typical individuals without pathology.
Response 1: Thank you for your words. We have conducted an exhaustive search of the bibliography on visual fixation in healthy patients and added a bibliography to complete the introduction and discussion. The changes have been added in highlighted red so they can be seen easier.